# Backwashing performance of self-cleaning screen filters in drip irrigation systems

**Quanli Zong** [1]*, **Zhenji Liu**[2]*, **Huanfang Liu**[2], **Hongfei Yang**[2]

**1** School of Resource and Environment, Qingdao Engineering Research Centre of Rural Environment, Qingdao Agricultural University, Qingdao, China, **2** College of Water Conservancy and Civil Engineering, Shihezi University, Shihezi, China

* quanli1871@126.com (QZ); 39080706@qq.com (ZL)

**Data Availability Statement:** All relevant data are within the manuscript.

**Funding:** This study was supported by the National Natural Science Foundation of China (Grant No.11662018 and No.51569029) and the Talent

## Abstract

The self-cleaning screen filter is one of the most common types used in drip irrigation systems. Backwashing pressure difference and backwashing time for two screen filters with one geometry and two different screens (178 μm and 124 μm) using two water qualities (tap water and sand–water mixture) were studied in a total of 88 runs (42 runs for tap water, 22 and 24 filtration cycles for sand–water mixture and backwashing, respectively). The backwashing pressure difference and backwashing time were calculated using the experimental data, and the results were largely in the range of measured values. Three constraint conditions (flowrate, sand condition and filtration time) of backwashing pressure difference were analysed, and the optimal values of backwashing pressure difference were given as 60.0 and 70.0 kPa for 178 μm and 124 μm filters, respectively. The backwashing time of the screen filter should be an optimal value that ensures that the pressure difference between the internal and external surfaces of the screen decreased to the initial value, and the sand concentration of the backwashed outlet decreased to a small, stable value. Based on the results of the backwashing experiment and prototype observation, the optimal backwashing time was given as 30 to 45 s for both screen filters.

## Introduction

Because emitter clogging is difficult to detect and the emitter is expensive to clean or replace, the clogging of emitters is the largest maintenance problem facing drip irrigation systems [1–3]. The suspended solids with both organic and inorganic components are the main reason for emitter clogging in drip irrigation However, the greatest clogging problems are caused by the presence of materials such as silt and algae [4–8]. Filtration is essential in the efficient operation of drip irrigation systems and also extends their operating life. Among all different types of filters, the automatic backwashing screen filter is one of the most common types used in drip irrigation systems [9].

The filter is the key component of drip irrigation, and several experimental studies have been performed on the performance of filters [9–10]. The performance of a filter can be defined by the head loss, the head loss ratio, the backwashing pressure difference, the

Research Startup Foundation of Qingdao
Agricultural University (663/1119031).

**Competing interests:** The authors have declared
that no competing interests exist.

backwashing time and the type of mechanical and hydraulic problems encountered [11–13]. Vieira et al developed and tested a novel concept for the filtration of particles in raw rainwater, in which the treatment efficiency for particle removal as well as the backwash efficiency were assessed for three different filter media [14].

Filter performance in drip irrigation systems has also been studied by a number of authors [9, 15–26]. Although some models and experiments results about the pressure loss of screen filters have been provided in previous studies, the results of backwashing performance, such as backwashing pressure difference and backwashing time, have not been provided.

Generally, screen filters are cleaned automatically using the backwashing system when the head loss reaches a fixed threshold or the clean period reaches an operation time. Although both options allow for easy system automation, automatic backwashing of filters may require a suitable or optimal backwashing pressure that the pumping system must supply, which is not the minimum or maximum. There are also some filters to be cleaned with automatic back-washing based on an operation time, regardless of whether the filter itself needs cleaning.

In Xinjiang Uygur Autonomous Region of China, self-cleaning screen filter is one of the most common types of filter. For ease of operation, the clean period of most self-cleaning screen filters is set to long fixed time. However, it will cause a large pressure difference between the inside and outside of screen surface, and the filter will undergo irreversible deformation destroy or incomplete cleaning, as shown in Fig 1. Therefore, the pressure difference is a good option to control the filter to be cleaned. When the pressure difference reached the pre-set value, the automatic backwashing system will work, and the filter will have to be cleaned to prevent screen from deformation destroy or incomplete cleaning.

The backwashing time is another important parameter to evaluate the effect of a backwashing system. Liu et al. and Li et al. have studied backwashing time through filter experiments [10, 27–28]. Duran et al. studied the performance of the screen, disc and a combination of screen and disc filters working with effluents used in micro irrigation systems and their

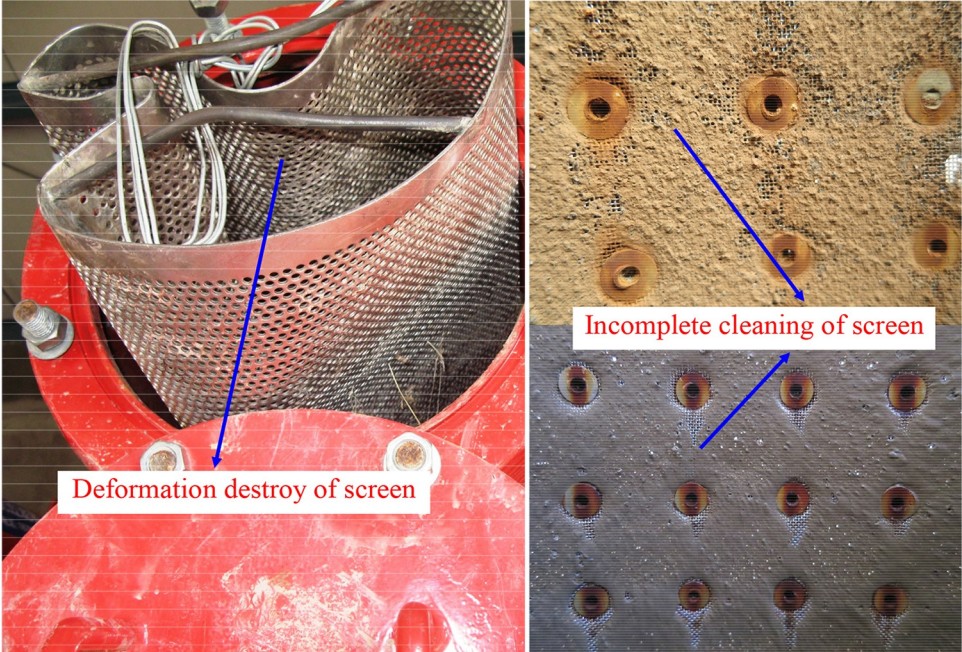

**Fig 1. Deformation destroy and incomplete cleaning of screen for self-cleaning screen filters.**

automatic backwashing efficiency at inlet pressures of 300 and 500 kPa [18]. It was concluded that the number of filters backwashing required for the screen filter was reduced at 500 kPa, and the disc filter at both 300 and 500 kPa consumed more water for backwashing than the screen filter and the combination of screen and disc filters.

Although the performance of filters commonly used in drip irrigation has been researched, there appears to be no analysis in the literature of the backwashing pressure difference and backwashing time in drip irrigation systems. The objective of this study is to analyse two parameters of automatic backwashing performance (backwashing pressure difference and backwashing time) of two screen filters with one geometry and two different screens (178 μm and 124 μm).

## Materials and methods

### Experimental set-up

In laboratory experiments, two different types of filters with one geometry and two screens were used to carry out the filtration experiments with the tap water and sand–water mixtures. The experimental filters were manufactured by the Shihezi "Jintudi" Water Saving Equipment Co., Ltd. of Xinjiang, China. Schematic representations of the screen filter used are given in Fig 2. The principle of self-cleaning screen filter can be found in Reference 9.

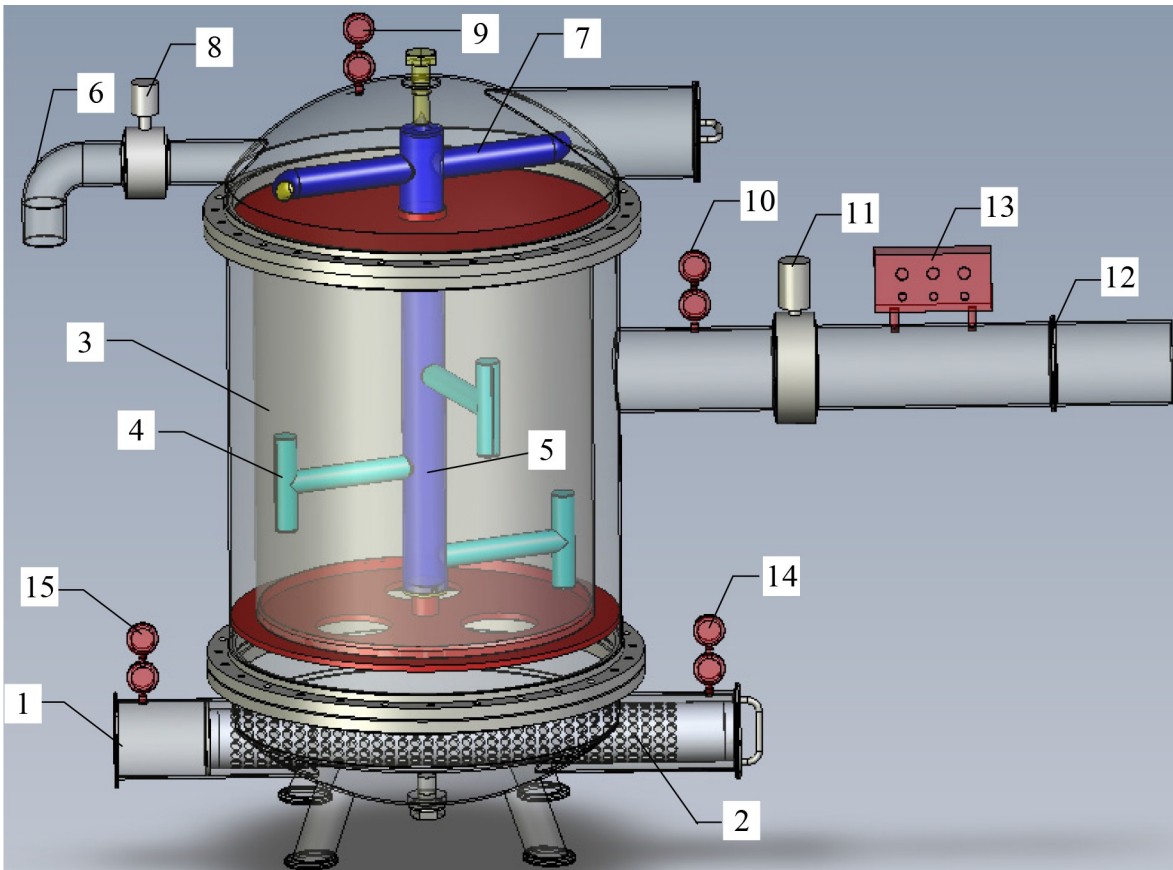

**Fig 2. Schematic representation of automatic backwashing screen filter for drip irrigation.** (1) Inlet; (2) first screen; (3) second screen; (4) suction tube; (5) flushing tube; (6) outlet of backwashing pipe; (7) driving pipe; (8) electromagnetic valve of backwashing pipe; (9) pressure gauge of backwashing; (10) pressure gauge of filtered water outlet; (11) electromagnetic valve of filtered water outlet pipe; (12) outlet of filtered water; (13) backwashing control system; (14) pressure gauge of second screen inlet; (15) pressure gauge of first screen inlet.

The data regarding various performance of filter, such as flowrates, sand concentration, filtration time and pressure loss was obtained by an experimental facility (Fig 3). A 30 m³ cuboid container (labelled (1) in Fig 3) was built to supply water, which the length, the width and height are 5.0 m, 4.0 m, and 1.5 m, respectively. A 1.131 m³ cylindrical mixing tank (labelled (2) in Fig 3) with the four-wing impeller of the mixer submerged to a depth of approximately 80% of the mixture's depth inside the tank, was also built to mix the water and sand particles mechanically, which an internal diameter and height were 1.2 m and 1.0 m, respectively.

Pressure losses at different flowrates were measured using two pressure gauges (MC03000128) with an accuracy of 0.2%, which were installed at the filter inlet and outlet. A frequency counter (labelled (7) in Fig 3) was installed to control the flowrates of experiments, which were measured using an ultrasonic flowmeter (TDS-100P) with an accuracy of 0.5%. The time for the backwashing experiments was recorded using a digital stopwatch. During the experiments, a digital thermometer was used to measure water temperature with values varied between 18 and 22°C. A centrifugal pump driven by a 1.5-kW motor (labelled (6) in Fig 3) was placed at inlet pipe of filter to make water be circulated.

## Operational procedure

Three types of experiments were conducted with two screen filters and two water qualities (tap water and sand–water mixture). For the tap water experiments, a total of 42 runs were tested, and the flowrates varied from 11.8 to 217.6 m³ h⁻¹. 19 and 23 experimental points were recorded for 178 μm and 124 μm filters, respectively. Table 1 showed the range of the main experimental variables.

In order to determine the relationships between the pressure loss and filtration time under different sand concentrations, sand–water mixture experiments were carried out following the tap water experiments. To begin each test series, sand was added until the desired concentration was attained within the system, which flowrate data were collected after the target values of sand concentration reached. The performances of the filters (pressure loss, filtration time, etc.) were measured for each sand concentration. Samples were taken after the data were acquired, and the sand addition phase was initiated. Filters were cleaned automatically using the backwashing system when the pressure loss reaching the filter exceeded the pre-set value of the pressure difference. To obtain the possible results of sand–water mixture experiment, the pre-set pressure difference of this experiment was set to a large value of 127.4kPa.

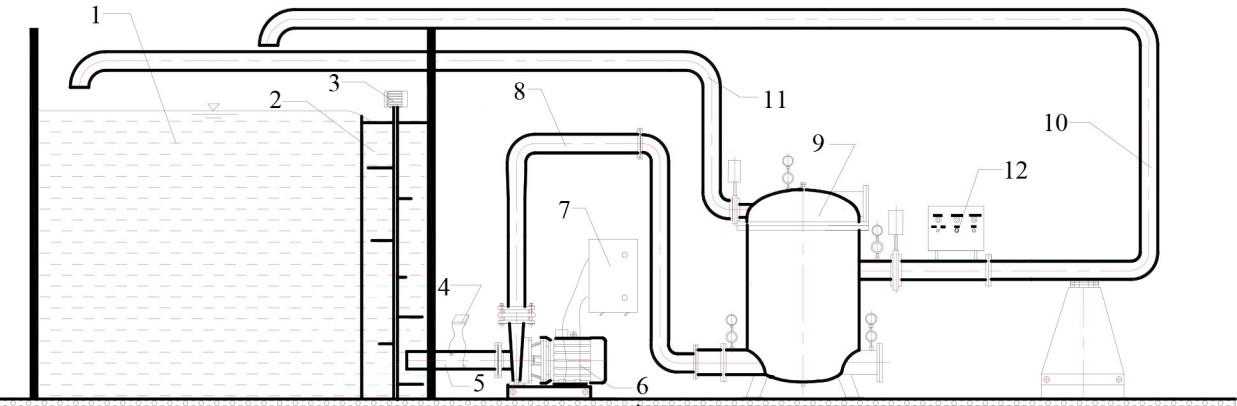

**Fig 3. Schematic representation of the test apparatus.** (1) cuboid container; (2) mixing tank; (3) mixer; (4) flowmeter; (5) inlet pipe of pump; (6) pump and motor; (7) frequency counter; (8) inlet pipe of filter; (9) screen filter; (10) filtered water pipe; (11) backwashing pipe; (12) backwashing control system.

**Table 1. Range (minimum–maximum values) of the main variables during the experiment.**

| Experiment type | Filtration level, μm | Number of runs | Minimum–maximum flowrate ($Q$), m³·h⁻¹ | Minimum–maximum sand concentration ($S$), kg.m⁻³ | Minimum–maximum pressure loss ($\triangle H$), kPa | Pre-set pressure difference for backwashing ($\triangle p$), kPa |
|---|---|---|---|---|---|---|
| Tap water | 178 | 19 | 15.5–217.6 | - | 0.93–45.08 | - |
| | 124 | 23 | 11.8–217.5 | - | 4.00–51.93 | - |
| Sand–water mixture | 178 | 12 | 217.6 | 0.103–0.193 | 45.08–131.30 | 127.4 |
| | 124 | 10 | 217.5 | 0.088–0.213 | 51.93–125.90 | 127.4 |
| Backwashing | 178 | 12 | 217.6 | 0.167–0.240 | - | 68.6, 78.4, 88.2, 98.0 |
| | 124 | 12 | 217.5 | 0.088–0.211 | - | 68.6, 78.4, 88.2, 98.0 |

During the backwashing experiment, the backwashing time was set to 100 s for both 124 μm and 178 μm screen filters, through which more data on the outlet sand concentration of the backwashing pipe can be obtained. To obtain the possible results of backwashing time under different pressure differences, the pre-set values of this experiment are 68.6, 78.4, 88.2 and 98.0 kPa, respectively. To analyse the physical parameters related to the backwashing time of the filter, samples were taken every 5 s at the outlet of the backwashing pipe. Thus, the relationship between the backwashing time and sand concentration of the outlet was recorded for each backwashing experiment.

Sand concentration of the inlet was varied from 0.088 to 0.213 kg m⁻³ and from 0.088 to 0.240 kg m⁻³ for sand–water mixture and backwashing experiments, respectively, as presented in Table 1. During the sand–water mixture experiment, 12 and 10 filtration cycles were carried out for 178 μm and 124 μm filters, respectively. For the backwashing experiment, there were 12 filtration cycles for each type of filter.

Fig 4 showed the particle size distribution of sand used in the experiments. As can be seen from Figure, the median particle sizes for the sand were 0.28 mm for 124 μm filter and 0.16 mm for 178 μm filter, which the measured average specific gravity was 2.63. The particle size distribution was tested using standard method of analysis by sieving. The sand samples had to be dried at 105°C before it was tested.

## Assessment of backwashing pressure difference

The screen filters may be defined based on filtration mechanisms, in which straining is the main filtration mechanism. Except that the suspended particles smaller than the size of mesh can pass through the mesh of screen, those particles larger than the mesh size will accumulate on the internal surface of the screen. Here, the suspended particle size is smaller than the mesh size, and the removal is dominated by physical mechanisms[4]. Thus, two processes may be defined on the basis of filtration mechanisms: (1) screen filtration and (2) filter cake filtration, as shown in Fig 5.

1. Screen filtration. As the water flows through the screen, the screen can block the particles that exceed the mesh size, allowing particles smaller than the mesh size to pass. In the initial stage, particles smaller than the mesh size pass through the screen with no blockage occurring. Particles larger than the mesh size in the water flow are blocked from passing through the mesh due to the inability to pass through the screen. The head of some particles can pass through the mesh, but the tail cannot pass through the mesh due to the irregularity of particle shape. Under the action of water pressure, these particles are embedded in the mesh of the screen, and forming a preliminary filter cake blockage together with particles larger than the mesh size.

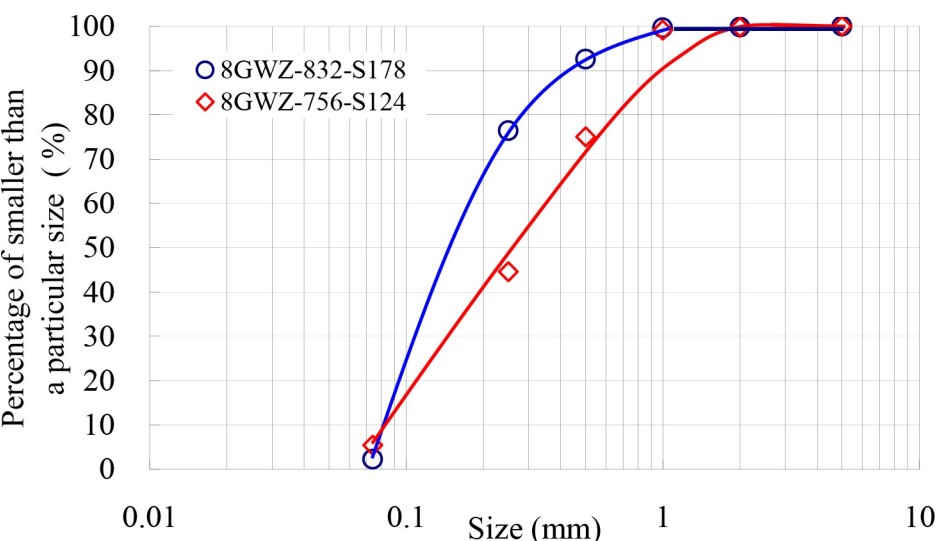

**Fig 4. Particle size distribution of sand used in the present study.**

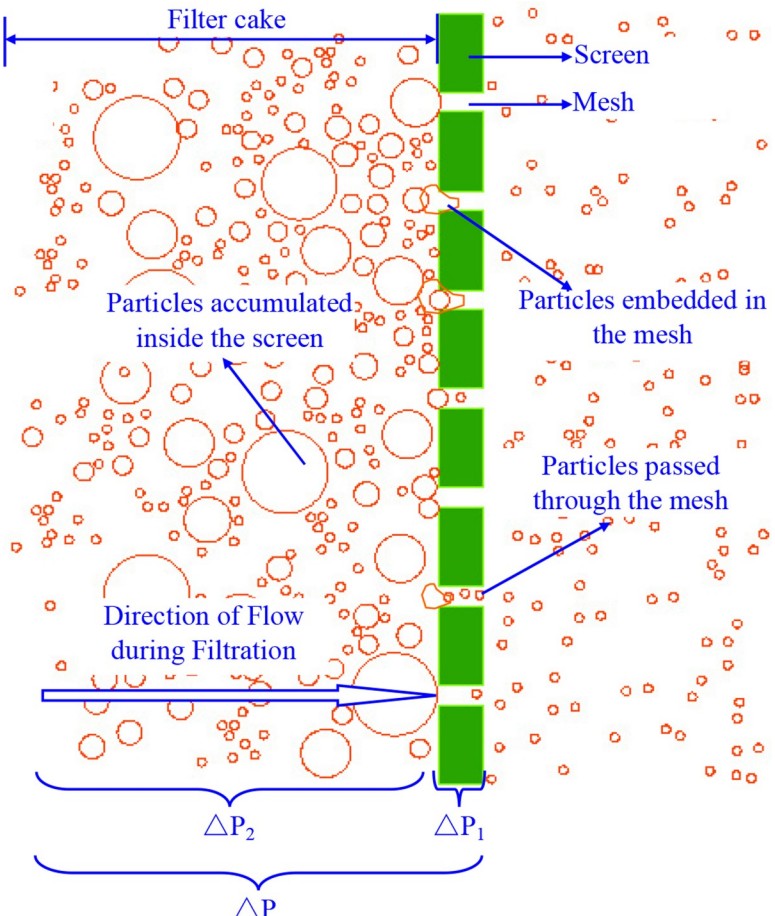

**Fig 5. Two processes based on the filtration mechanism.**

2. Filter cake filtration. As the particles embedded in the mesh of the screen and the particles retained on the surface of the screen are gradually stabilized by the interaction of the high-pressure and high-speed water flow and the adhesion between particles, the particles are difficult to move. As the particles gradually accumulate on the surface of the screen, they spread over the surface and gradually thicken, the screen blockage is intensified. The particles are self-assembled and stacked, and only a small gap or complicated passage inside allows the water flow or smaller particles to pass. Most of the particles are trapped, the ability to pass the water is reduced, and the porosity is reduced. Finally, the fine particles are also trapped. The pressure difference between the internal and external surface of the screen is continuously increased, and the pressure loss is increased to an unacceptable level. Filtration is difficult to proceed, and the filter cake blockage is formed.

As can be seen from Fig 5, the pressure loss of the screen filter can be calculated by two parts: (1) the pressure loss caused by the screen itself ($\triangle P_1$), in which the filter cake cannot be formed on the screens; and (2) the pressure loss caused by the cake formed on the screens ($\triangle P_2$).

Thus, the pressure loss of the screen filter can be expressed as Eq (1):

$$\Delta p = \Delta p_1 + \Delta p_2 \tag{1}$$

where $\triangle p$ is the pressure difference of the screen filter (Pa); $\triangle p_1$ is the pressure difference caused by the screen itself (Pa); and $\triangle p_2$ is the pressure difference caused by the filtration cake (Pa).

A mathematical expression of the pressure difference through a screen based on the filtration mechanism of straining is known as Darcy's law, which assumes the form

$$\Delta p_1 = \frac{\mu u L K_1 S_1{}^2 (1-\varepsilon)^2}{\varepsilon^3} \tag{2}$$

where $L$ is the thickness of media (m); $\mu$ is the water viscosity (Pa.s); $u$ is the stream velocity (m/s); $S_1$ is the single particle surface area (m$^2$); $\varepsilon$ is the porosity of media; and $K_1$ is the empirical coefficient, which is 5.0 when the porosity of the media $\varepsilon$ is between 0.3 and 0.5 [29].

Because Eq (2) was used only for laminar flow, the following formula provided by Heertjes et al. can be used to calculate the pressure loss for turbulent flow [30].

$$\Delta p_1{}^{1.55} = \frac{\mu u L K_1 S_1{}^2 (1-\varepsilon)^2}{\varepsilon^3} \tag{3}$$

Based on the diameter of mesh, the definition of the Reynolds number $Re$ was given by Heertjes et al. to determine the different modes of flow [30].

$$Re = \rho u d_p / \mu \tag{4}$$

where $Re$ is the Reynolds number flow through the mesh of the screen; $d_p$ is the diameter of the mesh (m); and $\rho$ is the water density in kg.m$^{-3}$.

According to calculated results of the Reynolds number, the modes of flow can be determined as follows: (1) when $Re$ is less than 3.0, it is laminar flow; (2) when $Re$ is greater than 7.0, it is turbulent flow; (3) when $Re$ is between 3.0 and 7.0, it is the transitive state [30].

It should be noted that an important cake parameter is its compressibility, which is particularly important when the rinsing of the filter is to be considered. Therefore, the pressure loss of cake is not easy to calculate. Because the filtration mechanism is the same as the screen itself

and cake, the pressure difference caused by the cake can be estimated by Eq (5) [31].

$$\Delta p_2 = \beta^3 \Delta p_1 \qquad (5)$$

where $\beta$ is an empirical coefficient between 2.5 and 4.0.

## Assessment of backwashing time

Filter backwashing consumes additional water, which is directly related to the backwashing time. Thus, the backwashing time is an interesting variable for micro-irrigation system design and management.

Several variables with some influence on the backwashing time of the screen filter have been previously identified [10,27–28]. The filter backwashing time can be determined by looking at the progressive decrease of the pressure difference across the screen. Under a certain flow rate and sand concentration of the filter inlet, the backwashing time should theoretically be the time in which the pressure difference between the internal and external surfaces of the screen decreased to the initial value at the beginning of filtration. However, because not all particles that accumulate on the internal surface of the screen can be backwashed, the pressure difference between the internal and external surfaces cannot be reduced to the initial value at the end of the backwashing processes; instead, it is slightly larger than the initial pressure difference.

Therefore, a parameter $P_m$ should be proposed, which is the ratio between the backwashed particles and the deposited particles on the surface of the internal screen. Thus, the end of the backwashing process is regarded as when the percentage of $P_m$ particles was backwashed.

According to the mass conservation of backwashed particles, the relationship regarding the backwashing time can be established. The total mass of particles deposited on the surface of the screen within the filtration time $t$ was

$$M = QSPt \qquad (6)$$

where $M$ is the total mass of particles deposited on the surface of the screen (kg); $Q$ is the flow-rate across the filter ($m^3 \ s^{-1}$); $S$ is the mean sand concentration in the filter inlet ($kg \ m^{-3}$); $P$ is the percentage of particle sizes greater than the mesh diameter of the screen; and $t$ is the filtration time (s).

The total mass of particles backwashed out of the filter within the backwashing time $t_p$ was

$$M_p = Q_p S_p t_p \qquad (7)$$

where $M_p$ is the total mass of particles backwashed out of the filter (kg); $Q_p$ is the flowrate of backwashing ($m^3 \ s^{-1}$); $S_p$ is the mean sand concentration in the backwashing outlet ($kg \ m^{-3}$); and $t_p$ is the backwashing time (s).

Based on the definition of the backwashing time and the mass conservation, the potential equation was

$$M_p = P_m \times M \qquad (8)$$

where $P_m$ is the ratio between the backwashed particles and the deposited particles on the surface of the screen.

By applying Eqs (6), (7) and (8), the following relationship can be established:

$$t_p = \frac{Q}{Q_p}\frac{S}{S_p}PP_m t \tag{9}$$

## Results and discussion

### Backwashing pressure difference of screen filter

**Calculated results of pressure difference of screen filter.** In this study, the medium of the screen is stainless steel metal mesh. The metal wire of the screen can be regarded as a smooth cylinder whose diameter is a constant, as shown in Fig 6. The diameters of the screen wire with different meshes were measured using Vernier callipers. For example, the diameters of the screen wire ($d$) were 103 μm and 71 μm for 178 μm and 124 μm screen filters, respectively. The total filtration surface was calculated from the screen area of the filter and the coefficient of the net area, which were 832,000 and 755,700 mm² for 178 μm and 124 μm screen filters, respectively [9].

The Reynolds number flow through the meshes of the screen can be calculated by substituting the standard values for the dynamic viscosity of water at 20°C as $1.005\times10^{-3}$ Pa s and the flow rate of 200 m³ h⁻¹. For example, the Reynolds number $Re$ was 11.8 and 9.1 for 178 μm and 124 μm screen filters, respectively. Therefore, the type of flow can be determined as turbulent flow. The pressure difference caused by the screen can be calculated by Eq (3).

Because the metal wire of the screen can be regarded as a smooth cylinder, the single particle surface area ($S_1$) can be defined as the surface per unit volume. Similarly, the porosity of media ($\varepsilon$) can be defined as the volume of the flow across the mesh, which can be expressed as

$$\varepsilon = \frac{\frac{1}{4}\pi d_p^{\,2}L}{(d_p + \mathrm{d})(d_p + \mathrm{d})L} = \frac{\pi d_p^{\,2}}{4(d_p + \mathrm{d})^2} \tag{10}$$

According to Eq (10), the porosity of media ($\varepsilon$) can be estimated as 0.315 and 0.317 for 178 μm and 124 μm screen filters, respectively.

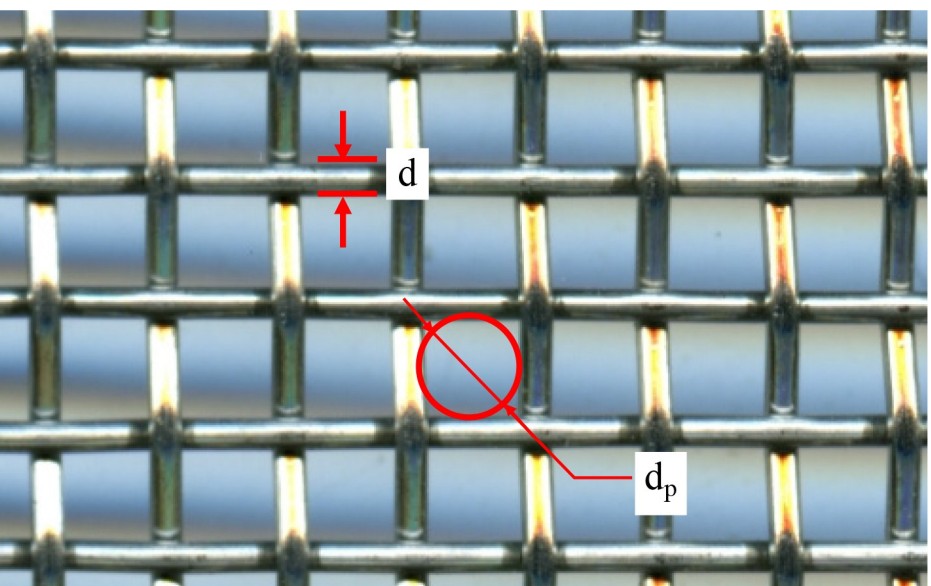

**Fig 6. Structure of the screen.**

By applying Eq (3), the pressure differences caused by the screen by substituting the value for the thickness of the medium with 200 μm were 1.31 and 2.21 kPa for 178 μm and 124 μm screen filters, respectively, as presented in Table 2. The calculated results of the pressure difference caused by the screen should be smaller than the pressure loss of tap water under the same flow because the latter also included the pressure loss caused by the pipelines and valves. For example, the measured pressure losses for tap water were 41.9 kPa and 43.9 kPa with a 200 m$^3$ h$^{-1}$ flowrate for 178 μm and 124 μm screen filters, respectively [9]; the calculated results of the pressure difference caused by the screen were both smaller than the measured values of tap water under the same flow.

By applying Eqs (3) and (5), the total pressure difference ranges of the screen filter can be calculated as 21.78 to 85.15 kPa and 36.74 to 143.65 kPa for 178 μm and 124 μm screen filters, respectively (Table 2). The calculated results of the pressure difference are basically in the ranges of the measured pressure losses for sand–water mixture with 23.5 to 131.3 kPa and 21.9 to 125.9 kPa. Although some calculated results exceeded the range of measured values, over-predictions are normal given that the calculated method makes predictions with a degree of error.

**Constraint conditions of backwashing pressure difference.** In the process of filtration, the flowrate and sand condition of the filter inlet have great influence on the pressure difference of the screen filter, which can determine the degree of clogging of the filter. In addition, the filtering process must maintain a certain time interval to ensure the normal work of irrigation systems instead of backwashing frequently. Therefore, the primary constraint conditions of backwashing pressure difference are flowrate, sand condition and filtration time.

It is obvious that the pressure loss of the filter increases with increasing flowrate. Under the worked flowrate of the filter, there should be a minimum backwashing pressure difference $\triangle p_{min}$, which is equal to the pressure loss of tap water with the worked flow rate. If the value of the pre-set backwashing pressure difference is smaller than this minimum backwashing pressure difference, the filter will not work properly and instead will be backwashing constantly.

The pressure losses of tap water for 178 μm and 124 μm screen filters were obtained by laboratory experiments with the regression coefficients $R^2$ are 0.951 and 0.956, respectively, as presented in Fig 7. As seen in Fig 7, the pressure losses were 45.08 kPa and 51.93 kPa under the 217.6 and 217.5 m$^3$ h$^{-1}$ flowrates for 178 μm and 124 μm screen filters, respectively. These two values of pressure losses should be the minimum backwashing pressure differences for 178 μm and 124 μm screen filters, respectively.

Under a certain flow, the influence of the sand condition on the backwashing pressure difference mainly includes the sand concentration and particle size. For example, the filter screen can be clogged quickly if the sand concentration is large, which can result in a rapid increase of pressure difference between the internal and external surface of the screen.

**Table 2. Calculated results of pressure difference for 178 μm and 124 μm screen filters.**

| Filtration level ,μm | The diameters of screen wire (d) ,μm | The screen area of the filter (A), mm² | The flowrate (Q), m³h⁻¹ | Reynolds number (Re) | The porosity of media (ε) | The pressure differences caused by the screen itself ($\triangle p_1$) ,kPa | The pressure difference caused by the filtration cake ($\triangle p_2$), kPa | | The pressure difference of the screen filter ($\triangle p$), Pa | |
|---|---|---|---|---|---|---|---|---|---|---|
| | | | | | | | $\beta = 2.5$ | $\beta = 4.0$ | $\beta = 2.5$ | $\beta = 4.0$ |
| 178 | 103 | 832,000 | 200 | 11.8 | 0.315 | 1.31 | 20.47 | 83.84 | 21.78 | 85.15 |
| 124 | 71 | 755,700 | 200 | 9.1 | 0.317 | 2.21 | 34.53 | 141.44 | 36.74 | 143.65 |

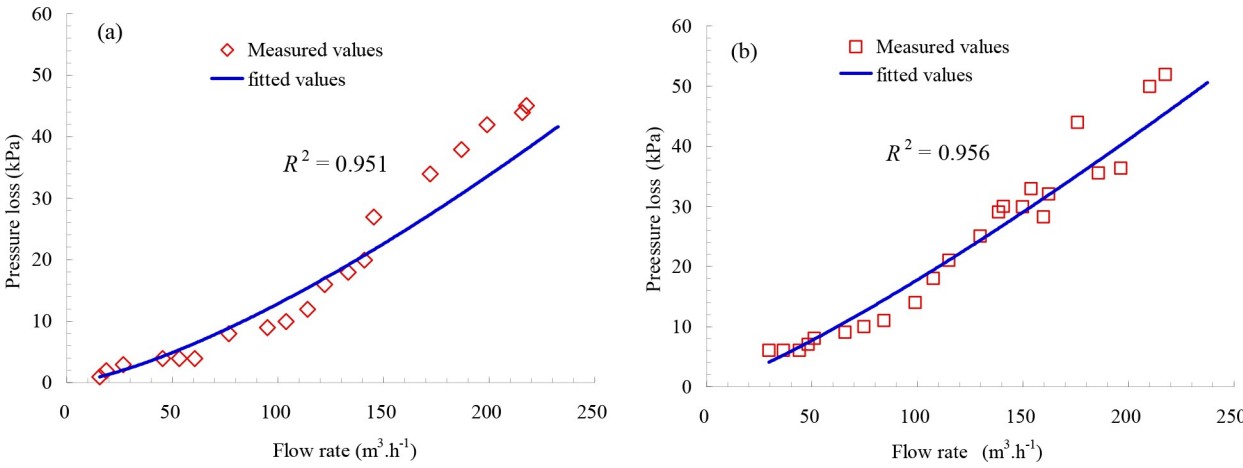

**Fig 7.** Backwashing pressure loss of the screen filter for tap water: (a) 178 μm; (b) 124 μm.

Fig 8 shows the relationship between filtration time and sand concentration with a 217.6 $m^3\ h^{-1}$ flow rate under different pressure losses for 178 μm filter, which the regression coefficients $R^2$ are 0.972, 0.915, 0.941, and 0.923 for the pressure losses of 49.0, 58.8, 78.4 and 98.0 kPa. As seen in Fig 8, under the same pressure loss, the filtration time decreased quickly with increasing sand concentration, which showed that the influence of sand concentration on the pressure loss is significant. However, the filter screen cannot be clogged in a short time if the particle size is small and most of them can pass through the screen, even if the sand concentration is relatively large. Conversely, the filter screen can be clogged in a short time if the particle size is large and most of them cannot pass through the screen, even if the sand

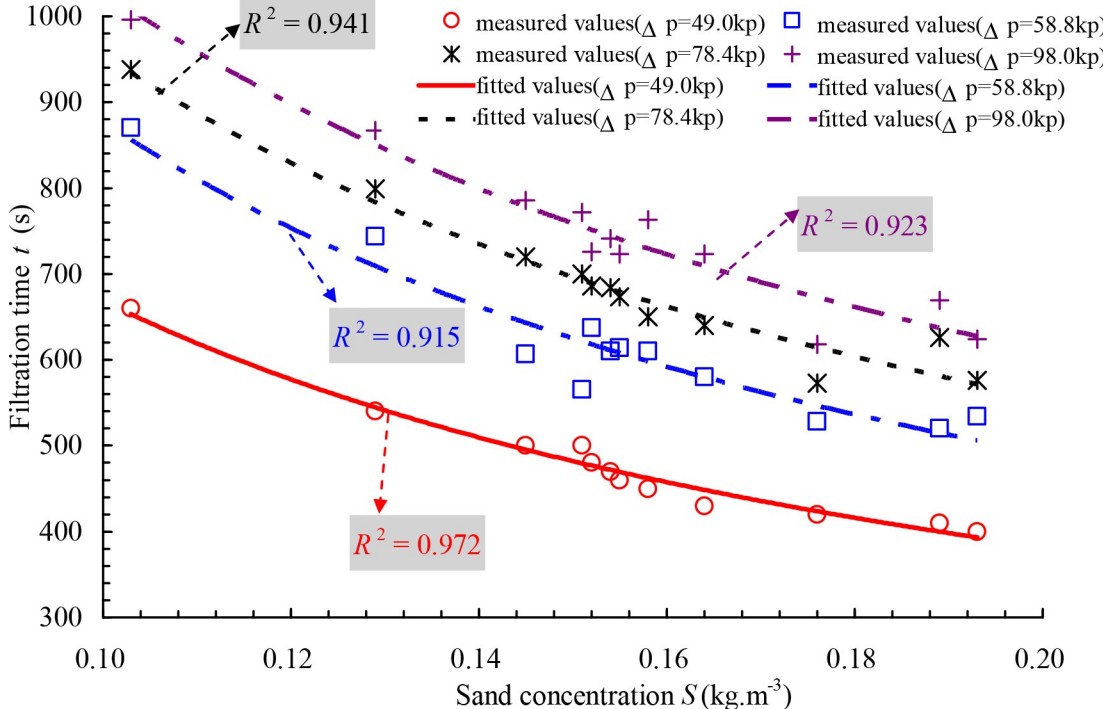

**Fig 8. Relationship between the filtration time and sand concentration under different pressure losses for 178 μm filter.**

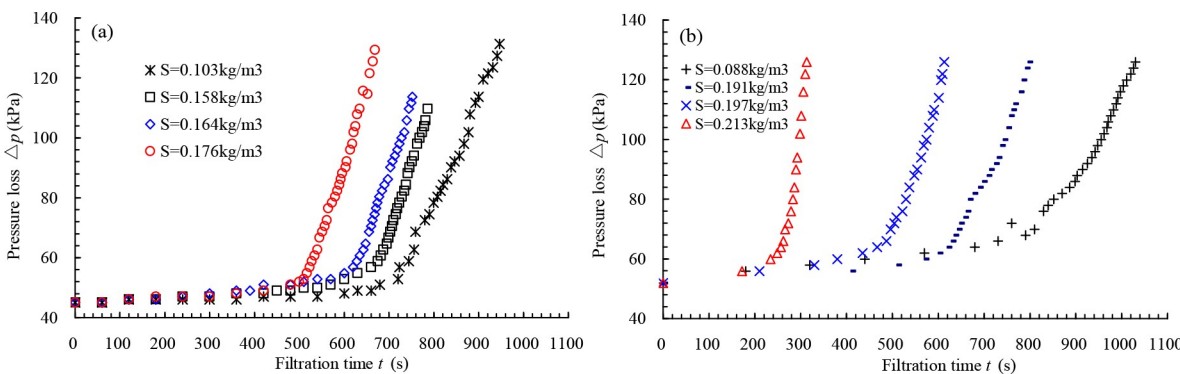

**Fig 9.** Relationship between the pressure loss and filtration time under different sand concentration with the same flowrate: (a) 178 μm filter with flowrate 217.6 m³/h; (b) 124 μm filter with flowrate 217.5 m³/h.

concentration is small. Therefore, the backwashing pressure difference should be determined by the sand condition of the filter inlet and the filtration time, considering that most meshes of the filter screen cannot be clogged, and the pressure loss of the filter was not increased sharply.

Under a certain sand condition, if the value of the backwashing pressure difference is set too small, the filtration time will be so short that the filter will be backwashed frequently. Conversely, if the value of the backwashing pressure difference is set too large, the filtration time will be prolonged but will lead to thickened deposited sand on the inner surface of the screen. Thus, some particles can be squeezed through the meshes of screen under the large pressure difference between the internal and external surfaces of the screen, which can enter the drip irrigation lines and clog the emitters. At the same time, the useful life of the screen will also decrease under the large pressure difference. Therefore, there should be a maximum pressure difference between the internal and external surfaces of the screen, which is the maximum backwashing pressure difference $\triangle p_{max}$.

Fig 9 shows the relationships between pressure losses and filtration times under the same flow rates with 217.6 and 217.5 m³/h for 178 μm and 124 μm filters, respectively. As seen in Fig 9, the pressure losses of filters increased gradually with increasing filtration time when the value of the pressure loss was within a threshold. However, the curve increases sharply when the pressure loss reaches the threshold, which shows that the screen of the filter was clogged quickly and needed to be backwashed.

**Optimal value of backwashing pressure difference.** It is concluded that the value of the backwashing pressure difference of the screen filter should be within a certain range, which is between the minimum ($\triangle p_{min}$) and maximum ($\triangle p_{max}$) backwashing pressure differences. Actually, there should be a threshold of pressure loss that ensures that the curve of the pressure loss does not increase sharply. Thus, this threshold is also the optimal value of the backwashing pressure difference.

As an example, for 178 μm screen filter, the range of pressure loss is 56.84 to 60.76 kPa with the curve of the pressure loss not to be increased sharply under different sand concentrations. For 124 μm screen filter, the range is 65.93 to 69.93 kPa. Thus, it can be concluded that the optimal values of backwashing pressure differences are 60.0 and 70.0 kPa for 178 and 124 μm filters, respectively.

To shorten the time of the filter experiment, a large median diameter sand was chosen that could clog the screen in a short time. Thus, the filtration times are very short under the above two optimal values of backwashing pressure differences, which are 947 and 1030 s for 178 μm and 124 μm filters, respectively.

However, in field irrigation, the sand basin was set to deposit the large sand particles, preventing them from entering the inlet of the filter. Therefore, most sand particles can pass through the meshes of the screen in an actual irrigation system. According to the prototype observation of the 143rd, 133rd and 144th groups of the Eighth division in the Xinjiang Production and Construction Corps, all sand particles entering the inlet of the filter are smaller than 0.18 mm, and the pressure loss increased slowly with increasing filtration time. The filtration time of all screen filters observed in field irrigation is at least several hours or ten hours with the optimal backwashing pressure differences being set to 60.0 and 70.0 kPa for 178 μm and 124 μm filters, respectively. Because the phenomenon of a filter being backwashed in a short time was not found, it is reasonable that the optimal backwashing pressure differences were set to the above values for 178 μm and 124 μm filters, respectively.

## Backwashing time

**Calculated results of backwashing time.** According to the particle size distribution of sand used in this study (Fig 4), the percentages of sizes greater than 178 μm and 124 μm are 34.0% and 76.0%, respectively, which are the values of $P$ in Eq (9). Thus, the backwashing times of different filters were calculated with the value of $P_m = 0.85$ in Eq (9) [28]. The relationship between the backwashing time $t_p$ and the product of the ratio of sand concentration between the inlet and backwashing outlet and filtration time $S/S_p \times t$ for 124 μm and 178 μm filters is shown in Fig 10. As seen in the Figure, the backwashing time increases with increasing product $S/S_p \times t$, and the relationship between them is nearly linear with the regression coefficients $R^2$ are 0.989 and 0.991 for 124 μm and 178 μm filters, respectively. When the sand concentration of the inlet and the filtration time for 178 μm filter were increased from 0.103 to 0.193 kg.m$^{-3}$ and from 679 to 1038 s, respectively, the backwashing time was increased from 18 to 39 s with a mean value of 31 s. For 124 μm filter, the calculated backwashing time was from 26 to 67 s and the mean value is 56 s with the range of the inlet sand concentration and the filtration time from 0.088 to 0.213 kg.m$^{-3}$ and from 1030 to 313 s, respectively. The reason for the difference of the mean backwashing time between 178 μm and 124 μm filters is that the

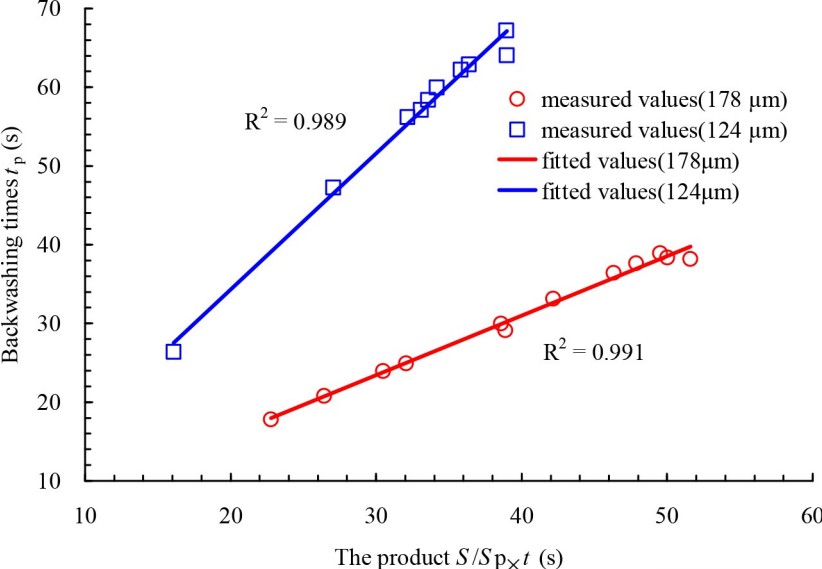

**Fig 10. Relationship between the backwashing time $t_p$ and the product of the ratio of sand concentration between the inlet and backwashing outlet and filtration time $S/S_p t$ for 124 μm and 178 μm filters.**

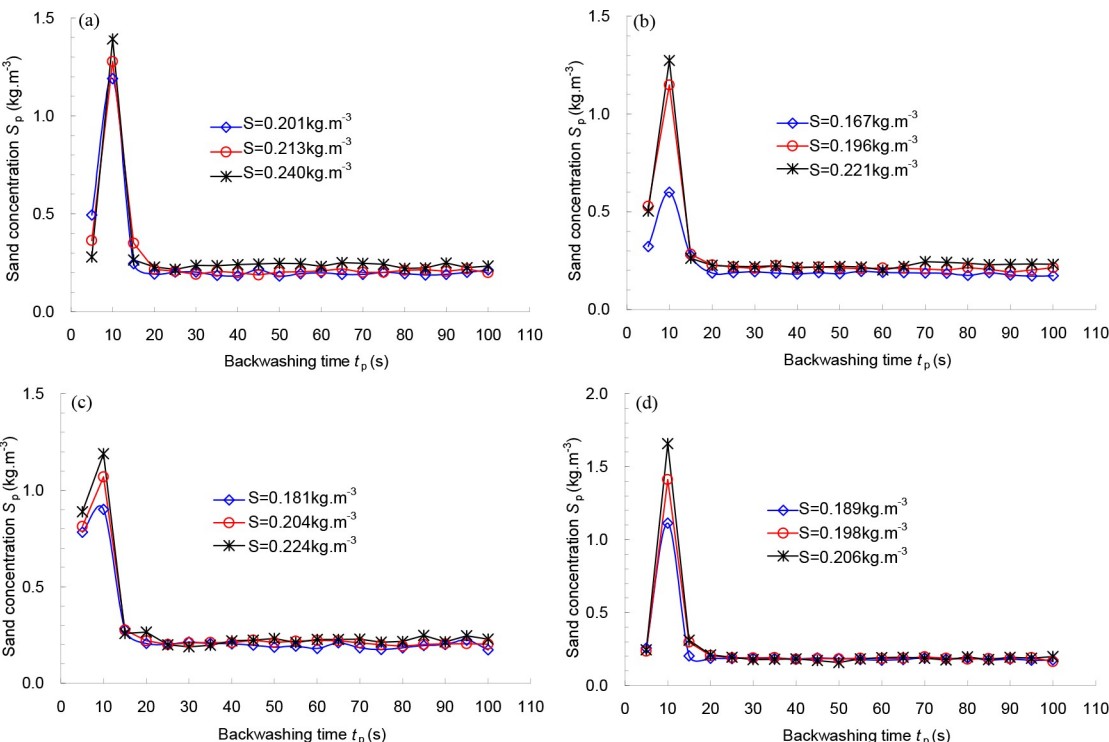

**Fig 11.** Relationship between the backwashing time $t_p$ and sand concentration of backwashed outlet $S_p$ under different sand concentrations of the inlet for 178 μm filter with pre-set backwashed pressure differences of (a) 68.6 kPa; (b) 78.4 kPa; (c) 88.2 kPa; (d) 98.0 kPa.

values of $P$, which represent the percentage of particle size greater than that of screen mesh size. The mean value of calculated backwashing time is 42 s for both filters.

**Optimal value of backwashing time.** Because the filter backwashing time can be determined by looking at the progressive decrease in the pressure difference across the screen, there should be a minimum value in which the pressure difference between the internal and external surfaces of the screen decreased to the initial value. In other words, the sand concentration of the backwashing outlet should be reduced to a small, stable value at the end of backwashing processes. At the same time, because filter backwashing consumes additional water, there also should be a maximum backwashing time. Therefore, there should be a threshold of backwashing time that ensures that the pressure difference between the internal and external surfaces of the screen decreased to the initial value, and the sand concentration of the backwashed outlet decreased to a small, stable value. Thus, this threshold is also the optimal value of the backwashing time.

Fig 11 and Fig 12 show the relationships between the backwashing time $t_p$ and sand concentration of the backwashed outlet $S_p$ under different sand concentrations of the inlet for 178 μm and 124 μm filters, respectively. As seen in both figures, under a certain flowrate and pre-set backwashing pressure difference, close relationships exist between the backwashing time and sand concentration of the backwashed outlet. With increasing backwashing time, the sand concentration of the backwashing outlet first increases and then decreases until it reaches a steady value with this threshold for the minimum backwashing time. For 178 μm filter, the range of minimum values of the backwashing time under different pre-set backwashing pressure differences is from 20 to 30 s (see Fig 11). For 124 μm filter, the minimum value is

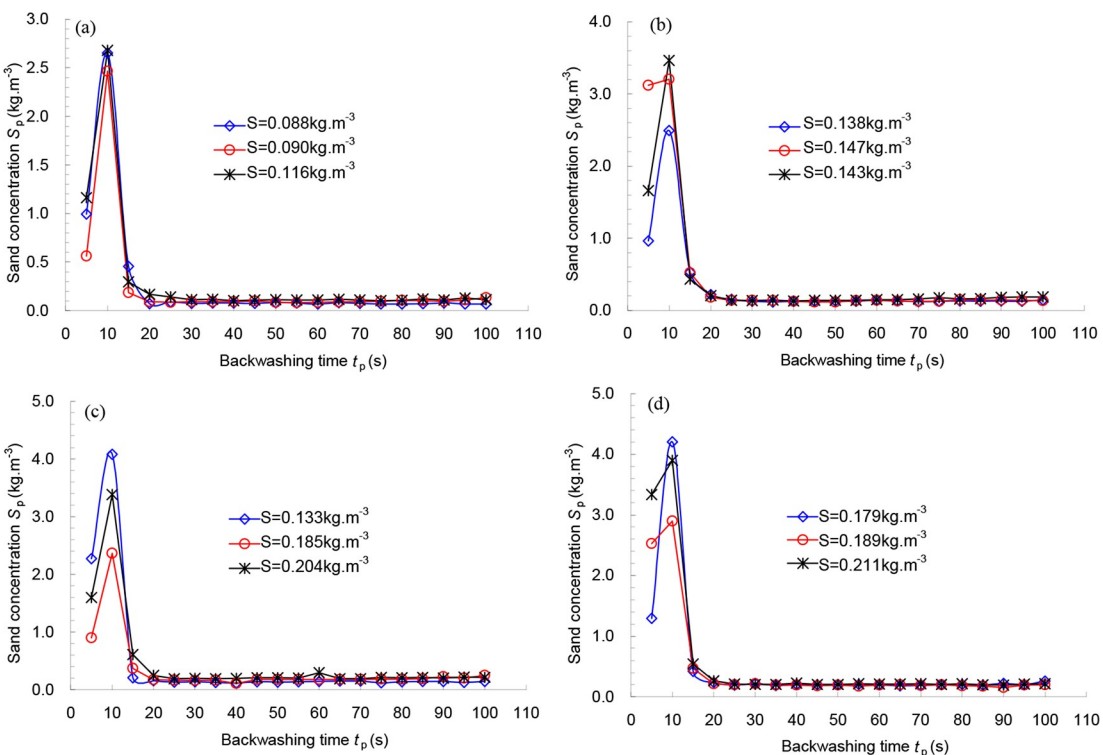

**Fig 12.** Relationship between the backwashing time $t_p$ and sand concentration of backwashed outlet $S_p$ under different sand concentrations of the inlet for 124 μm filter with pre-set backwashed pressure differences of (a) 68.6k Pa; (b) 78.4 kPa; (c) 88.2 kPa; (d) 98.0 kPa.

approximately 20 s (see Fig 12). All curves for the sand concentration of the backwashing outlet can reach a steady value after the backwashing time reaches 30 s.

To ensure the efficiency of backwashing and stable work for the next filtration cycle, the backwashing time of most screen filters in field irrigation was set to 30 to 45 s. For example, all screen filters used in the 143rd, 133rd and 144th groups of the Eighth division in the Xinjiang Production and Construction Corps worked well with the backwashing time set to 30 to 45 s. The pressure differences observed in field irrigation between the internal and external surfaces of the screens can also be reduced to the initial values after the filters were backwashed. Therefore, it is reasonable that the optimal backwashing time was set to 30 to 45 s for both screen filters, which is basically consistent with the calculated mean backwashing time of 42 s.

## Conclusions

An experiment with two screen filters and two water qualities (tap water and sand–water mixture) was performed. Data obtained from the filtration and backwashing tests allowed two parameters of the backwashing performance (backwashing pressure difference and backwashing time) of the screen filters to be computed. Although some calculated results exceeded the range of measured values, the calculated values of backwashing pressure difference of 21.78 to 85.15 kPa and 36.74 to 143.65 kPa for 178 μm and 124 μm screen filters, respectively, and mean backwashing time of 42 s are basically in the range of the measured values.

Three constraint conditions influencing the backwashing pressure difference are incorporated into the flowrate, sand condition and the filtration time, which determine the minimum and maximum values of the backwashing pressure difference. The minimum backwashing pressure difference was the pressure loss of tap water with maximum working flowrates (217.6

and 217.5 m$^3$h$^{-1}$), which are 45.08 kPa and 51.93 kPa for 178 μm and 124μm screen filters, respectively. The maximum backwashing pressure difference was the threshold value between the internal and external surfaces of the screen, which ensures that the curve of the pressure loss does not increase sharply, with ranges of pressure loss from 56.84 to 60.76 kPa and from 65.93 to 69.93 kPa for 178 μm and 124 μm filters, respectively. Thus, the optimal values of backwashing pressure differences can be determined as 60.0 and 70.0 kPa for 178 μm and 124 μm filters, respectively, which are proved to be reasonable by the prototype observation results of the 143[rd], 133[rd] and 144[th] groups of the Eighth division in the Xinjiang Production and Construction Corps, China.

The backwashing time of two screen filters working under four different pre-set pressure differences, with average values of sand concentration between 0.088 and 0.240 kg.m$^{-3}$ and flowrates 217.6 and 217.5 m$^3$h$^{-1}$, was affected by the flowrates of the filter inlet and backwashing outlet, the mean sand concentration in the filter inlet and backwashing outlet, the percentage of particle sizes greater than the mesh diameter of the screen, the ratio between the backwashed particles and the deposited particles on the surface of the screen, and the filtration time. The optimal backwashing time is the threshold that ensures that the pressure difference between the internal and external surfaces of the screen was reduced to the initial value, and the sand concentration of the backwashed outlet was reduced to a small, stable value. Based on the results of the backwashing experiment and prototype observation, the optimal backwashing time was given as 30 to 45 s for both screen filters. The pressure differences observed in field irrigation between the internal and external surfaces of the screens can be reduced to the initial values after the filters were backwashed with the optimal backwashing time.

## Acknowledgments

The authors would like to express their gratitude to the National Natural Science Foundation of China (Grant No.11662018). The authors would also like to thank the Shihezi "Jintudi" Water Saving Equipment Co., Ltd of Xinjiang, China for its help in developing this experiment at their installations. Thanks are due to Fei Liu and Xiuping Luo for their help in carrying out the experiments.

## Author Contributions

**Data curation:** Zhenji Liu, Hongfei Yang.

**Writing – original draft:** Quanli Zong.

**Writing – review & editing:** Quanli Zong, Huanfang Liu.

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
