## [Decision Letter · Decision Letter 0]

22 Oct 2019

PONE-D-19-26474

Backwas h ing Performance o f Self-cleaning Screen Filters in Drip Irrigation Systems

PLOS ONE

Dear Dr Zong,

Thank you for submitting your manuscript to PLOS ONE. After careful consideration, we feel that it has merit but does not fully meet PLOS ONE’s publication criteria as it currently stands. Therefore, we invite you to submit a revised version of the manuscript that addresses the points raised during the review process.

We would appreciate receiving your revised manuscript by Dec 06 2019 11:59PM. To enhance the reproducibility of your results, we recommend that if applicable you deposit your laboratory protocols in protocols.io, where a protocol can be assigned its own identifier (DOI) such that it can be cited independently in the future. For instructions see: http://journals.plos.org/plosone/s/submission-guidelines#loc-laboratory-protocols

We look forward to receiving your revised manuscript.

Kind regards,

Vassilis G. Aschonitis

Academic Editor

PLOS ONE

**Journal Requirements**

http://www.journals.plos.org/plosone/s/file?id=wjVg/PLOSOne_formatting_sample_main_body.pdf and http://www.journals.plos.org/plosone/s/file?id=ba62/PLOSOne_formatting_sample_title_authors_affiliations.pdfReviewer's Responses to Questions

**Comments to the Author**

1. Is the manuscript technically sound, and do the data support the conclusions?

Reviewer #1: Yes

2. Has the statistical analysis been performed appropriately and rigorously? 

Reviewer #1: Yes

3. Have the authors made all data underlying the findings in their manuscript fully available?

Reviewer #1: Yes

4. Is the manuscript presented in an intelligible fashion and written in standard English?

Reviewer #1: Yes

5. Review Comments to the Author

Reviewer #1: This is new contribution in the area of drip irrigation and adds new knowledge in filtering systems.

You may delete all introductory text leaving only findings and chapter objectives.

Insert Figures and tables close to the text

6. PLOS authors have the option to publish the peer review history of their article (what does this mean?). If published, this will include your full peer review and any attached files.

Reviewer #1: No

---

## [Author Response · Author response to Decision Letter 0]

22 Nov 2019

The authors appreciate the efforts made by the reviewers and the Editor in providing the following comments and suggestions. We have addressed all of the comments. Revised portion are marked in red in the Revised Manuscript with Track Changes.

---

## [Editor Report · Decision Letter 1]

26 Nov 2019

Backwashing Performance of Self-cleaning Screen Filters in Drip Irrigation Systems

PONE-D-19-26474R1

Dear Dr. Zong,

We are pleased to inform you that your manuscript has been judged scientifically suitable for publication and will be formally accepted for publication once it complies with all outstanding technical requirements.

With kind regards,

Vassilis G. Aschonitis

Academic Editor

PLOS ONE
---

## [Editor Report · Acceptance letter]

10 Dec 2019

PONE-D-19-26474R1 

Backwashing Performance of Self-cleaning Screen Filters in Drip Irrigation Systems 

Dear Dr. Zong:

I am pleased to inform you that your manuscript has been deemed suitable for publication in PLOS ONE. Congratulations! Your manuscript is now with our production department. 

With kind regards,

on behalf of

Dr. Vassilis G. Aschonitis 

Academic Editor

PLOS ONE